# Examining the psychometric properties of the Controlling Coach Behaviors Scale in Chinese Elite Athletes

**Daliang Zhao, Yu Zhou** [ID]*

School of Leisure Sport and Management, Guangzhou Sport University, Guangzhou, China

* 1045256659@qq.com

**Data Availability Statement:** https://osf.io/6rwts/.

**Funding:** 21省社科共建项目 72201880228 Role of Funder statement: the funder acted as the first author in study. Funding [21省社科共建项目 72201880228].

## Abstract

Coaching style is key to athletes' performance and mental well-being. However, few attempts have examined the effects of coaching style on athletes in a Chinese cultural context. Based on previous literature and 23 expert interviews (11 with athletes and 12 with coaches), we rebuilt a 16-items on the Controlling Coach Behaviors Scale. In study 1, 130 provincial team athletes completed a 16-items questionnaire on controlling coach behaviors. The questionnaire items were then screened using exploratory factor analysis and transformed into a formal scale. In study 2, another 560 provincial athletes completed several measures related to coaching style, motivation, subjective vitality, and burnout, and systematic tests were carried out to validate the scale. Study 3 examined the test-retest reliability of the Controlling Coach Behaviors Scale over a 2-week interval. Finally, the present study yielded a nine-item Controlling Coach Behaviors Scale with three dimensions (controlling use of reward, negative conditional regard, and excessive personal control). It suggests that cultural differences played an important role in the communication between athletes and coaches. The new Controlling Coach Behaviors Scale shows good validity and can be used in future research.

## Introduction

Coaches play an essential role in athletes' psychological experiences as well as their athletic performance [1]. The study of coaching styles originated in self-determination theory (SDT) [2,3]. SDT holds that environmental factors condition whether an individual's psychological needs are satisfied or blocked. For athletes, coaches' attitudes and behavioural tendencies are crucial environmental factors. Researchers interested in coaching styles have applied the SDT framework in various ways. Currently, the most widely accepted view is to divide coaching styles into "autonomy-supportive" and "controlling" [4–8]. The autonomy-supportive coaching style typically manifests as active support from coaches that encourages spontaneous efforts from athletes and creates the conditions in which athletes can experience a sense of will, choice, and identity [9–11]. In contrast, the controlling coaching style imposes a specific, preconceived way of thinking and behaving on athletes in a coercive and authoritarian way [12,13].

**Competing interests:** The authors have declared that no competing interests exist.

**Table 1. Types of controlling coach behaviours.**

| Type | Interpretation | Example |
|---|---|---|
| 1. Rewards | Use of external rewards | My coach tries to motivate me by promising to reward me if I do well. |
| 2.Negative conditional regard | Responding to athletes' failure with indifference in the hope that the athlete will work harder and achieve better results in the future | My coach pays me less attention if I displease him/her. |
| 3. Intimidation | Humiliating and belittling the athlete or threatening to use corporal punishment | My coach shouts at me in front of others to make me do certain things. |
| 4.Excessive personal control | Intervening invasively in athletes' lives outside of the training environment | My coach tries to control what I do in my free time. |

Source: The types of behaviours are drawn from the Controlling Coach Behaviors Scale developed by Bartholomew et al. (2010) [14].

Bartholomew et al [14] identified four dimensions of the controlling coaching style: *controlling use of rewards*, *intimidation*, *negative conditional regard*, and *excessive personal control* (see Table 1). The *controlling use of rewards* is the use of external rewards to control behaviour; *negative conditional regard* is responding to athletes' failure with indifference in the hope that the athlete will work harder and achieve better results in the future; *intimidation* is humiliating and belittling the athlete or threatening to use corporal punishment; and *excessive personal control* is intervening invasively in athletes' lives outside training [14,15]. Bartholomew et al [14] developed a scale that was subsequently used to explore the effects of the controlling coaching style on other psychological variables [16–19]. Controlling coaching was found to hinder the improvement of athletic performance via adverse psychological outcomes (e.g., mental fatigue, eating disorders, and negative emotions) [14].

Zhao et al [15] tested the validity of the scale with Chinese athletes. They found that it had three deficiencies that limited the scale's usability. First, there were no cross-factor loading tests and the scale dimensions had not been confirmed. It was unclear whether these same four dimensions (*controlling use of rewards*, *negative conditional regard*, *intimidation*, and *excessive personal control*) were suitable for the characteristics of Chinese athletes. Second, the test only contained two items measuring *excessive personal control*, which reduced the validity of the scale. Third, there was instability in the external validity of the "*controlling use of rewards*" measure. While Bartholomew et al [14] found a significant negative correlation between *controlling use of rewards* and autonomy support, Zhao et al [15] observed that *controlling use of rewards* was not significantly correlated with autonomy support or subjective vitality. The inconsistencies between these two studies may be due to cultural differences.

In the context of Chinese culture, many athletes leave their families and parents at an early age and live and train in sports teams for long periods of time. They are with their coaches day and night and form special coach–athlete relationships. Traditional Chinese culture, characterised by proverbs such as 'a teacher for a day, a father for a lifetime' and 'a filial son will be born under a stick, and a strict teacher will create a good student', may lead to differences in Chinese and Western athletes' feelings about controlling coaches. This study hypothesises that athletes in Chinese contexts have different feelings about controlling coaching styles than Western athletes. Therefore, it is necessary to explore the external validity of the Controlling Coach Behaviors Scale and develop a scale that is suitable for evaluating coaching styles in Chinese contexts. Drawing on Bartholomew et al [14] and Zhao et al [15], we formed an item pool through interviews. Study 1 examined the factor loadings of each item using exploratory factor analysis. In Study 2, confirmatory factor analysis was used to test the scale's structural stability, convergent

validity, divergent validity and internal consistency. Study 3 examined the test-retest reliability of the scale.

## Study 2

Study 1 was designed to generate an item pool based on the results of the interviews, drawing heavily on the work of Zhao et al [15] and Bartholomew et al [14]. Exploratory factor analysis was carried out to examine the scale's dimensions.

### Method

**Compilation and revision of the item pool.** *Pre-interviews*. A semi-structured interview outline containing 14 questions was developed around the theme of how coaches interact with athletes. The pre-interview participants were two coaches and four athletes. Following each pre-interview, the research team revised the interview outline by reviewing the interview recording and discussing it. The outline was revised six times (once per interview). Finally, three core topics were determined based on the characteristics of the coaches and athletes. Items in the coach interview outline included: "What methods do you use to get athletes to achieve your goals or intentions?"; "How do you exercise control over athletes during training?", and "Talk about the relationship between you and your athletes." Items in the athlete interview outline included: "What methods does your coach use to get you to achieve his or her goals or intentions?"; "How does your coach exercise control over athletes during training?", and "Talk about the relationship between you and your coach."

*Formal interviews*. The study used a purposeful sampling method in which 11 athletes (seven males and four females; $M_{age}$ = 14.5, $SD$ = 2.1) and 12 coaches (ten males and two females; $M_{age}$ = 36.2, $SD$ = 3.4) from the G Provincial Team were invited to participate in face-to-face interviews. All of these athletes had over 5 years of training experience and specialised in one of the following five sports: martial arts, gymnastics, table tennis, diving, and volleyball. The coaches had all been coaching for more than 4 years and specialised in one of the following eight sports: martial arts, gymnastics, swimming, weightlifting, table tennis, badminton, volleyball, and basketball. The recording procedure and confidentiality guidelines were explained to interviewees before the interviews and informed consent was obtained. The duration of the interviews was between 40 and 80 minutes. To maintain consistency, all of the interviews were conducted by the same person.

*Analysis*. Nvivo 11.0 software was used to analyse the interview results. Two researchers transcribed the interview recordings verbatim and then imported them into the software. The results of the discussions, along with the results of subsequent statistical analyses, yielded an item pool for our Controlling Coach Behaviors Scale for Chinese Athletes. The *controlling use of rewards* dimension used three items from the original scale, the *intimidation* dimension used four items from the original scale, and the *negative conditional regard* dimension also used four items from the original scale. The *excessive personal control* dimension incorporated three new items and the two items in the original scale, for a total of five items. The final item pool contained 16 items. The researchers followed the translation-back translation method to translate the Controlling Coach Behaviors Scale from English to Chinese. The specific procedure was as follows. Two bilingual (Chinese–English) translators independently translated the questionnaire into Chinese and then discussed the translated questionnaire until they reached a consensus, thus forming a preliminary Chinese questionnaire. Then, another bilingual translator independently translated the preliminary Chinese questionnaire into English. The researchers compared the translated English questionnaire with the original English

questionnaire to test the accuracy of the translation. The Chinese version of the questionnaire was finalized through discussion and modification.

**Participants.** The participants were 130 provincial team athletes ($M_{age}$ = 18.59, $SD$ = 3.63; $M_{training\ years}$ = 8.58, $SD$ = 3.84) who participated in sports such as martial arts, *sanda* (Chinese kickboxing), table tennis, and weightlifting. All procedures performed in this study have been approved by the Institutional Review Board of the Guangzhou Sport University. All participants or their parents included in the study1, study2 and study3 signed informed consent.

**Procedure.** The Controlling Coach Behaviors Scale with 16 items was adopted and generated based on the results of the interviews, drawing heavily on the work of Zhao et al [15] and Bartholomew et al [14].

**Data analysis.** SPSS 20.0 was used for the data analysis. Each item was evaluated using exploratory factor analysis. Items with a factor loading of below 0.5 were deleted and no cross-loadings of items were allowed.

## Results

**Project analysis.** The participants' total scores on the questionnaire were ordered from highest to lowest. The *high* group consisted of people who scored in the top 27%, while the *low* group consisted of people who scored in the bottom 27%. The difference between the scores of the high group and the low group on each item was calculated using an independent-samples *t*-test. The differences between the high and low groups on all of the items were found to be statistically significant ($t(68)$ = 2.46, $p \leq .017$).

The correlation between each item and the total score was then calculated. All of these correlation coefficients were found to be significantly greater than 0.4 ($r(70)$ = .36, $p < .001$).

**Exploratory factor analysis.** Exploratory factor analysis and extraction were conducted using principal component analysis. We used a varimax rotation to conduct an exploratory factor analysis of this 16-item Controlling Coach Behaviors Scale. After several rounds of item deletion, nine items were retained. The sample suitability test (KMO = 0.73) and spherical test results ($\chi^2$ = 557.10, $p < .001$) showed that the sample was suitable for factor analysis (see Table 2).

**Table 2. Item means, standard deviation, factor loadings, and skewness and kurtosis value following exploratory factory analyses (Study 1).**

| Item | M | SD | Excessive Personal Control | Negative Conditional Regard | Controlling Use of Rewards | Skewness | Kurtosis |
|---|---|---|---|---|---|---|---|
| The only reason my coach rewards/praise me is to make me train harder. | 3.72 | 1.6 | | | 0.83 | -0.24 | 0.19 |
| My coach tries to motivate me by promising to reward me if I do well. | 3.25 | 1.46 | | | 0.83 | -0.01 | -0.43 |
| My coach only uses reward/praise so that I stay focused on task during training. | 3.48 | 1.45 | | | 0.87 | 0.05 | -0.26 |
| My coach is less supportive of me when I am not training and completing well. | 3.3 | 1.61 | | 0.85 | | 0.85 | -0.45 |
| My coach pays me less attention if I have displeased him/her. | 3.47 | 1.62 | | 0.8 | | -0.18 | -0.71 |
| My coach is less accepting of me if I have disappointed him/her. | 3.62 | 1.61 | | 0.88 | | -0.13 | -0.20 |
| My coach tries to interfere in aspects of my life outside of my sport. | 2.94 | 1.46 | 0.81 | | | 0.20 | -0.99 |
| My coaches tried to control everything I did. | 2.54 | 1.41 | 0.87 | | | 0.82 | 0.31 |
| My coach tries to control what I do during my free time. | 2.98 | 1.46 | 0.74 | | | 0.30 | -0.53 |

Three co-factors were sampled, with a cumulative contribution rate of 75.03. The loading of each item on each factor was between 0.74 and 0.88. All four items in the *intimidation* dimension and one item in the *negative conditional regard* dimension were deleted due to cross-factor loadings. The *excessive personal control* dimension contained three items. The *controlling use of rewards* dimension retained its original three items. In summary, our Chinese Controlling Coach Behaviors Scale contained three dimensions (*controlling use of rewards*, *negative conditional regard*, and *excessive personal control*), and each of these dimensions contained three items.

The eigenvalue for *negative conditional regard* was 3.742, accounting for 41.58% of the interpretable variance. The eigenvalue for *excessive personal control* was 1.72, accounting for 19.15% of the interpretable variance. The eigenvalue of *controlling use of rewards* was 1.29, accounting for 14.30% of the interpretable variance.

## Study 2

The purpose of study 2 was to conduct a confirmatory factor analysis of the Controlling Coach Behaviors Scale, test whether the division into these three dimensions was reasonable, and test the scale's external validity. The study draws on Schumann's 7-point guide as a general framework [20]. Accordingly, molecular (skew check) and molar (reliability) preconditions are basically fulfilled. The questionnaires have been validated by previous research as sufficiently reliable (study) and also discriminates from other non-related concepts.

### Method

**Participants.**   The participants were 560 athletes (290 males and 270 females; $M_{age}$ = 18.68, $SD$ = 3.95; $M_{training years}$ = 7.99, $SD$ = 4.12) from provincial sports teams. They represented the following sports: swimming, weightlifting, gymnastics, basketball, volleyball, martial arts, *sanda*, table tennis, athletics, badminton, culminal swimming, water polo, and tennis.

**Measures.**   *The Controlling Coach Behaviors Scale for Chinese Athletes*. This scale, which we developed in study 1, had nine items and three dimensions. The *controlling use of rewards* dimension consisted of three items, including "The only reason my coach rewards or praises me is to make me train harder." The *negative conditional regard* dimension consisted of three items, including "My coach is less supportive of me when I am not training and competing well." The *excessive personal control* dimension consisted of three items, including "My coach tries to interfere in aspects of my life outside of my sport." A 5-point Likert scale was used, ranging from 1 (*strongly disagree*) to 5 (*strongly agree*).

*The Athlete Motivation Scale*. The Athlete Motivation Scale, compiled by Pelletier et al [21], had 18 items arranged into six dimensions. The amotivated regulation dimension contained three items, such as "I used to have good reasons for doing sports, but now I am asking myself if I should continue." The external regulation dimension contained three items, such as "I participate in my sport because people around me reward me when I do." The introjected regulation dimension contained three items, such as "I participate in my sport because I feel better about myself when I do." The identified regulation dimension contained three items, such as "I participate in my sport because I have found it is a good way to develop aspects of myself that I value." The integrated regulation dimension contained three items, such as "I participate in my sport because it is consistent with my personal values." The intrinsic regulation dimension contained three items, such as "For the pleasure I feel in living exciting experiences." A 7-point Likert scale was used, with responses ranging from 1 (*do not agree at all*) to 7 (*completely agree*). Li et al [22] translated the items into Chinese and found that the Athlete Motivation Scale still had good validity when used with Chinese athletes ($\varphi$ = .85 -.94).

*The Autonomy Support Scale.* The six-item Autonomy Support Scale [23] was adapted from the health field to measure the degree to which athletes perceived that their coaches supported their autonomy. This instrument used a 7-point Likert scale ranging from 1 (*strongly disagree*) to 7 (*strongly agree*). The Chinese version of this scale has shown good reliability and validity in the field of exercise [24], with an internal consistency value of 0.882.

*The Subjective Vitality Scale.* A six-item version of the Subjective Vitality Scale [25] was used to measure the athletes' subjective vitality. This instrument uses a 7-point Likert scale ranging from 1 (*strongly disagree*) to 7 (*strongly agree*). This scale has shown good reliability and validity in other Chinese populations [26], with an internal consistency value of 0.830.

*The Burnout Scale.* Burnout symptoms were assessed using the Athlete Burnout Questionnaire (ABQ) [27]. This instrument has 15 items on three subscales: an emotional and physical exhaustion subscale, which includes items such as "I am exhausted by the mental and physical demands of my sport"; a reduced accomplishment subscale, which includes items such as "I am not achieving much in my sport"; and the sport devaluation subscale, which includes items such as "I have negative feelings toward my sport." A 5-point Likert scale was used, ranging from 1 (*almost never*) to 5 (*almost always*). Research has confirmed the reliability [28], factorial validity [27], and convergent and divergent validity [29] of ABQ scores.

**Data analysis.** SPSS 20.0 and AMOS 22.0 were used for the data analysis. We used confirmatory factor analysis to assess the structural validity of the Controlling Coaching Behaviors Scale. The following parameters were used to determine whether the model showed a good fit: $\chi^2/df \leq 3$, RMSEA $\leq .08$ [30], CFI $\geq .95$, TLI $\geq .95$ [31], PNFI $\geq .50$ [32]. A correlation analysis was conducted to test the validity.

## Results

**Confirmatory factor analysis.** Population parameters are displayed in Table 2–1. A confirmatory factor analysis was performed using the three dimensions (*controlling use of rewards*, *negative conditional regard*, and *excessive personal control*) as potential variables. The results showed that the model fitted the data well ($\chi^2/df = 2.770$, RMSEA = .057, CFI = .985, TLI = .977, PNFI = .651; see Fig 1).

To exclude the possibility that controlling coaching style had a single dimension, all of the items were used as observation variables and a single submersible variable was used to model the structural equations. The relevant indicators were as follows: $\chi^2/df = 43.128$, RMSEA = .275, CFI = .596, TLI = .461, PNFI = .443. The correction according to the result of goodness of fit was not acceptable. The model fit did not meet acceptable standards, confirming that the Controlling Coach Behaviors Scale had a multidimensional structure composed of multiple control strategies (see Table 2–2).

The possibility that there was a single dimension in the Controlling Coach Behaviors Scale was therefore excluded. However, in the three-factor model, the correlation between *controlling use of rewards* and *negative conditional regard* was low, as was the correlation between *controlling use of rewards* and *excessive personal control*. No *controlling use of rewards* dimension was included in previous scales measuring controlling coaching [33,34]. We temporarily

**Table 2–1. Population parameters.**

| | | Study1 | | | | Study 2 | | | | Study 3 | | | |
|---|---|---|---|---|---|---|---|---|---|---|---|---|---|
| | Measures | M | SD | Skew | Range | M | SD | Skew | Range | M | SD | Skew | Range |
| 1 | Controlling Use of Rewards | 3.256 | 1.140 | -.007 | 1–6 | 3.558 | 1.403 | .339 | 1–7 | 3.062 | 1.429 | .884 | 1–7 |
| 2 | Negative Conditional Regard | 3.236 | 1.197 | -.179 | 1–6 | 3.292 | 1.388 | .232 | 1–7 | 2.749 | 1.157 | .317 | 1–6 |
| 3 | Excessive Personal Control | 2.577 | 1.075 | .164 | 1–5 | 2.756 | 1.345 | .810 | 1–7 | 2.128 | .804 | .137 | 1–5 |

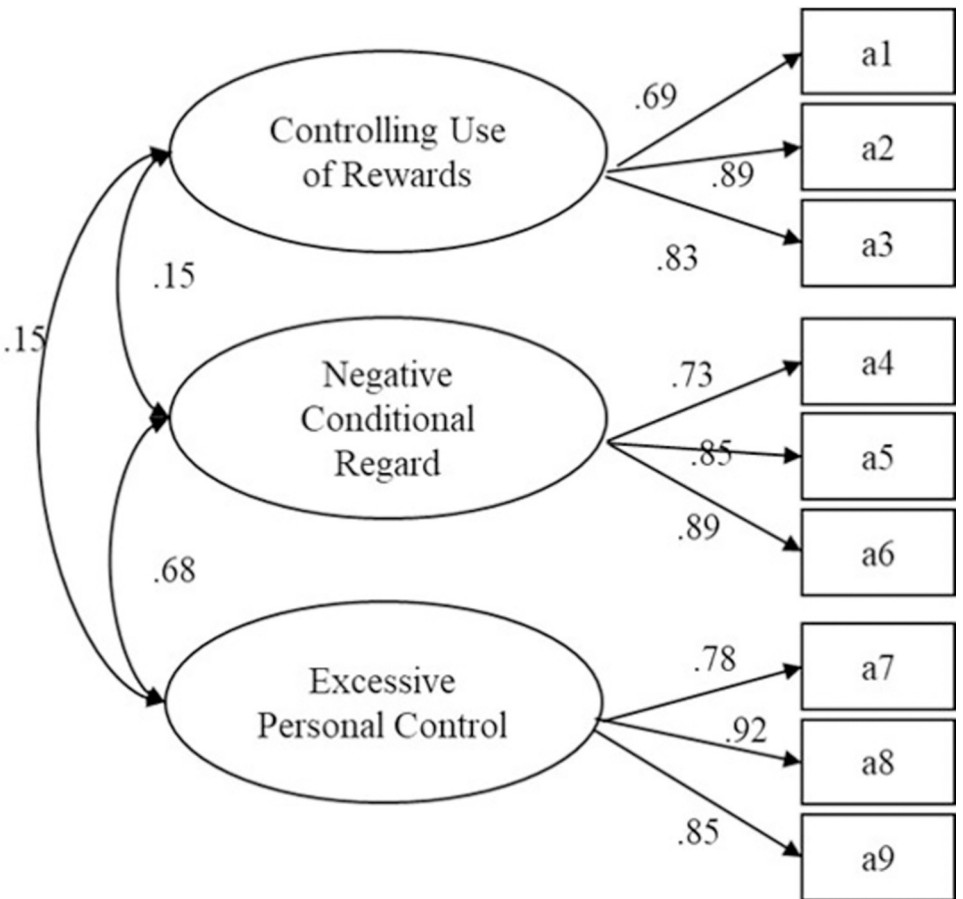

**Fig 1. Results of three-factor confirmatory factor analysis of the Controlling Coach Behaviors Scale.**

deleted the *controlling use of rewards* dimension to determine the fit of a two-dimensional model. Confirmatory factor analysis was performed using two dimensions (*negative conditional regard* and *excessive personal control*) as potential variables and the results showed that the model fit well ($\chi^2$/df = 4.635, RMSEA = .081, CFI = .986, TLI = .974, PNFI = .524; see Table 2–2 and Fig 2).

Finally, the results of a first-order confirmatory factor analysis showed a high correlation ($r = 0.68$) between the two underlying variables of *negative conditional regard* and *excessive personal control*, suggesting that the model may have a high-level factor structure. A second-order two-factor confirmatory factor analysis model was therefore constructed. Using ML estimation to fit the model, the main fit indicators were as follows: $\chi^2 = 0$, RMSEA = .497, CFI = 1, TLI = 0, PNFI = 0. The results showed that the model did not reach the ideal state. It suggested that the model did not have a high-level factor structure.

**Table 2–2. Multidimensional dimension and single dimension fitting indicators.**

| | $\chi^2$/df | RMSEA | CFI | TLI | PNFI |
|---|---|---|---|---|---|
| three-dimensional model | 2.770 | .057 | .985 | .977 | .651 |
| two-dimensional model | 1.425 | .028 | .999 | .997 | .999 |
| one-dimensional model | 43.128 | .275 | .596 | .461 | .443 |

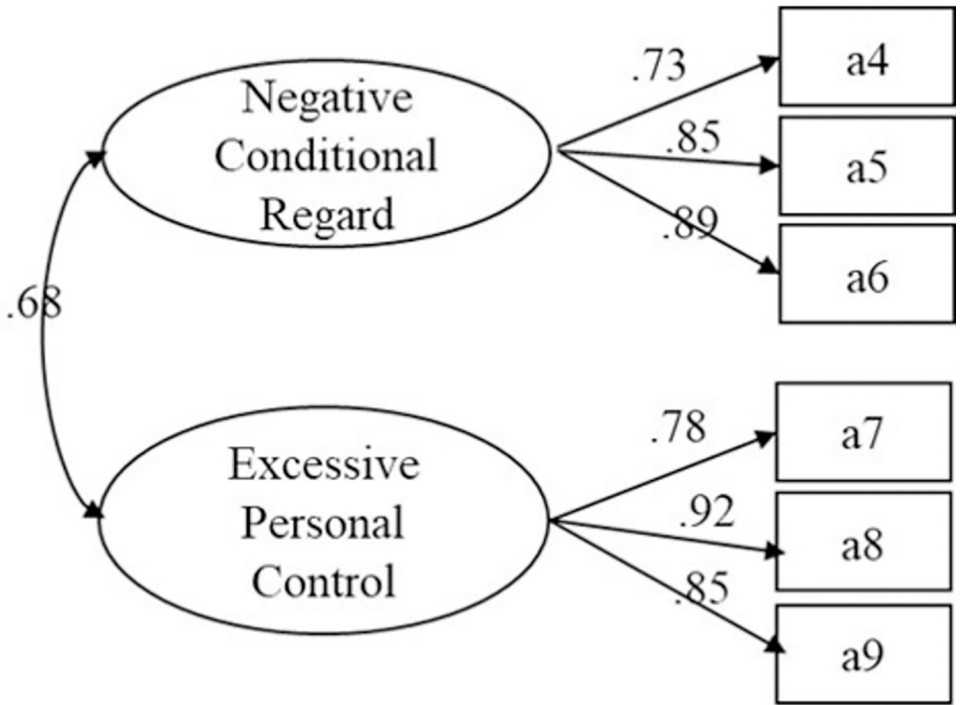

**Fig 2. Results of two-factor confirmatory factor analysis of the Controlling Coach Behaviors Scale.**

**Stability of structure.** To verify the stability of the scale's structure, the study analysed male and female participants separately. In the female condition, a confirmatory factor analysis was performed using all three dimensions (rewards, negative conditional regard and excessive personal control) as potential variables. The results showed that the model fit the data well ($\chi2/df = 1.637$, RMSEA = .049, CFI = .989, TLI = .983, PNFI = .648; see Table 2–3). For the potential variables, a confirmatory factor analysis was performed using two dimensions (negative conditional regard and excessive personal control). The results of the goodness of fit tests were as follows: $\chi2/df = 3.160$, RMSEA = .090, CFI = .984, TLI = .969, PNFI = .521. The adjusted model fit values were as follows: $\chi2/df = 3.160$, RMSEA = .037, CFI = .998, TLI = .995 and PNFI = .397, indicating a good fit (see Table 2–3).

In the male condition, a confirmatory factor analysis was performed using the three dimensions (controlling use of rewards, negative conditional regard and excessive personal control) as potential variables. The results of the goodness of fit tests were as follows: $\chi2/df = 2.562$,

**Table 2–3. Model structure fitting under different conditions.**

|  |  | χ2/df | RMSEA | CFI | TLI | NFI |
|---|---|---|---|---|---|---|
| Female | three dimensions | 1.637 | .049 | .989 | .983 | .648 |
|  | Two dimensions | 3.160 | .037 | .998 | .995 | .397 |
| male | three dimensions | 2.562 | .044 | .991 | .986 | .520 |
|  | Two dimensions | 1.175 | .025 | .999 | .997 | .992 |
| Closed sport | three dimensions | 1.760 | .050 | .989 | .983 | .649 |
|  | Two dimensions | 3.130 | .061 | .994 | .986 | .461 |
| Open sport | three dimensions | 2.768 | .047 | .990 | .983 | .568 |
|  | Two dimensions | 3.018 | .089 | .981 | .965 | .972 |

RMSEA = .074, CFI = .974, TLI = .961, PNFI = .639. The adjusted model fit values were as follows: $\chi2/df$ = 2.562, RMSEA = .044, CFI = .991, TLI = .986 and PNFI = .597. For the potential variables, a confirmatory factor analysis was performed using two dimensions (negative conditional regard and excessive personal control). The results of the goodness of fit tests were as follows: $\chi2/df$ = 3.285, RMSEA = .089, CFI = .982, TLI = .967 and PNFI = .520. The adjusted model fit values were as follows: $\chi2/df$ = 3.285, RMSEA < 0.001, CFI = 1.000, TLI = 1.007 and PNFI = .399, indicating a good model fit (see Table 2–3).

In addition, we divided the sports into two categories: closed sports and open sports. A confirmatory factor analysis was performed on the closed sport subsample using three dimensions (controlling use of rewards, negative conditional regard and excessive personal control) as potential variables. The results showed a good model fit ($\chi2/df$ = 1.760, RMSEA = .050, CFI = .989, TLI = .983 and PNFI = .649). For the potential variables, a confirmatory factor analysis was performed using two dimensions (negative conditional regard and excessive personal control). The results for the goodness of fit tests were as follows: $\chi2/df$ = 3.130, RMSEA = .084, CFI = .986, TLI = .974 and PNFI = .523. The adjusted model fit values were as follows: $\chi2/df$ = 3.130, RMSEA = .061, CFI = .994, TLI = .986 and PNFI = .461, indicating a good model fit (see Table 2–3).

A confirmatory factor analysis was performed on the open sports subsample, using three dimensions (controlling use of rewards, negative conditional regard and excessive personal control) as potential variables. The results of the goodness of fit tests were as follows: $\chi2/df$ = 2.768, RMSEA = .083, CFI = .965, TLI = .948 and PNFI = .631. The adjusted fit values were as follows: $\chi2/df$ = 2.768, RMSEA = .047, CFI = .990, TLI = .983 and PNFI = .568. For the potential variables, a confirmatory factor analysis was performed using two dimensions (negative conditional regard and excessive personal control). The results of the goodness of fit tests were as follows: $\chi2/df$ = 3.018, RMSEA = .089, CFI = .981, TLI = .965 and PNFI = .519. The adjusted goodness of fit values were as follows: $\chi2/df$ = 3.018, RMSEA < 0.001, CFI = 1.000, TLI = 1.008 and PNFI = .398, indicating a good model fit (see Table 2–3).

**Correlational structure.** The correlation analysis showed that autonomy support was negatively and significantly correlated with *negative conditional regard* and *excessive personal control*. Autonomy support was positively and significantly correlated with *controlling use of rewards*. Subjective vitality was negatively and significantly correlated with *negative conditional regard* and *excessive personal control* and positively and significantly correlated with *controlling use of reward*. Reducing the sense of value was positively and significantly correlated with *controlling use of rewards*, *negative conditional regard*, and *excessive personal control*. Decreased sense of achievement was positively and significantly correlated with *negative conditional regard* and *excessive personal control* and not significantly related to *controlling use of rewards*. Emotional and physical depletion was positively and significantly correlated with *negative conditional regard* and *excessive personal control* but not significantly associated with *controlling use of reward*. External regulation showed a positive and significant correlation with the *controlling use of rewards*, *negative conditional regard*, and *excessive personal control*. Intrinsic regulation was positively and significantly correlated with *controlling use of rewards*, *negative conditional regard*, and *excessive personal control*. *Controlling use of rewards* was positively correlated with intrinsic, identified, and integrated regulation.

To analyse the stability of the divergent and convergent validity of the scale, 560 subjects were randomly divided into two groups. Sample 1 contained 280 subjects, and Sample 2 contained 280 subjects. The relationships between the three dimensions and the other variables were consistent in all of the samples, indicating that the scale's divergent and convergent validity had strong stability (see Table 2–4).

**Table 2–4. The convergent vs. divergent validity of Controlling Coach Behaviors Scale.**

| Variable | Controlling Use of Rewards | | | Negative Conditional Regard | | | Excessive Personal Control | | |
|---|---|---|---|---|---|---|---|---|---|
| | Total | Sample 1 | Sample2 | Total | Sample 1 | Sample2 | Total | Sample 1 | Sample2 |
| Autonomy supportive | .192** | .207** | .160** | -.471** | -.415** | -.523** | -.373** | -.323** | -.441** |
| Subjective vitality | .228** | .235** | .198** | -.274** | -.310** | -.222** | -.243** | -.266** | -.221** |
| Diminished sense of worth | .093* | .123* | 0.086 | .415** | .448** | .375** | .347** | .403** | .297** |
| Diminished personal Accomplishment | -0.082 | -0.087 | -0.072 | .377** | .332** | .420** | .252** | .252** | .255** |
| Emotional body exhaustion | -0.012 | 0.019 | -0.022 | .382** | .427** | .325** | .338** | .411** | .261** |
| Intrinsic | .249** | .259** | .213** | -.253** | -.225** | -.270** | -.250** | -.175** | -.355** |
| External | .230** | .160** | .310** | .314** | .357** | .276** | .355** | .380** | .327** |
| Introjected | .174** | .204** | .147* | .155** | .226** | 0.08 | 0.03 | .138* | -.099* |
| Amotivation | 0.13 | .118* | .157* | .454** | .464** | .443** | .408** | .448** | .363** |
| Identified | .151** | .199** | 0.073 | -.223** | -.228** | -.207** | -.241** | -.199** | -.299** |
| Integrated | .192** | .240** | 0.114 | -.191** | -.147** | -.224** | -.181** | -0.09 | -.302** |

Note. * for p<0.05

** for p<0.01.

**Internal consistency.** Cronbach's α for the whole Controlling Coach Behaviors Scale was 0.819, indicating that the internal consistency of the scale was good. The threshold of internal consistency of each dimension was [0.819–0.881] (see Table 2–5).

## Study 3

Study 3 examined the test-retest reliability of the Controlled Coaching Style Scale by conducting a pre-test 2 weeks before a post-test, using a sample of 44 athletes.

### Method

**Participants.** The participants were 44 athletes (16 males and 28 females; $M_{age}$ = 20.45, $SD$ = 3.87; $M_{training\ years}$ = 10.64, $SD$ = 4.42) from provincial sports teams. They participated in weightlifting and wushu sports.

**Measures.** *The Controlling Coach Behaviors Scale for Chinese Athletes*. This scale, which we developed in study 1, had nine items and three dimensions. The *controlling use of rewards* dimension consisted of three items, including "The only reason my coach rewards or praises me is to make me train harder." The *negative conditional regard* dimension consisted of three items, including "My coach is less supportive of me when I am not training and competing well." The *excessive personal control* dimension consisted of three items, including "My coach tries to interfere in aspects of my life outside of my sport." A 5-point Likert scale was used, ranging from 1 (*strongly disagree*) to 5 (*strongly agree*).

### Result

**Retest reliability.** The reliability coefficients are shown along the main diagonal of the correlation matrix (Table 3–1). As expected, the total score indicated high reliability ($r$ = 0.918,

**Table 2–5. The reliability of Controlling Coach Behaviors Scale.**

| Total α | Controlling Use of Rewards | Negative Conditional Regard | Excessive Personal Control |
|---|---|---|---|
| .819 | .840 | .864 | .881 |

**Table 3–1. The test-retest reliability of Controlling Coach Behaviors Scale.**

| | | Second test | | | |
|---|---|---|---|---|---|
| **First test** | | **1** | **2** | **3** | **4** |
| Controlling Use of Rewards | 1 | .837** | | | |
| Negative Conditional Regard | 2 | | .945** | | |
| Excessive Personal Control | 3 | | | .881** | |
| Total | 4 | | | | 918** |

**Note.** ** $p < 0.01$
* $p < 0.05$.

95% *CI* [.856, .962]), and good reliability scores were obtained for Controlling Use of Rewards (*r* = 0.837, 95% *CI* [.629, .926), Negative Conditional Regard (*r* = 0.945, 95% *CI* [.893, .979]) and Excessive Personal Control (*r* = 0.881, 95% *CI* [.813, .931]). We conducted a post-hoc power analysis (i.e., sample size = 44, number of observations = 2, *p* = .05 and effect size = .70) using PASS 11 and found that there was enough statistical power (0.99) to conduct the intra-class correlation tests [22]. In summary, the responses for the Controlling Coach Behaviors Scale had reliable stability.

## Discussion

The Controlling Coach Behaviors Scale, compiled by Bartholomew et al [14], includes four dimensions: *controlling use of rewards* (four items), *intimidation* (four items), *negative conditional regard* (four items), and *excessive personal control* (three items), for a total of 15 items. Zhao et al [15] conducted a preliminary test of the validity of this scale on two independent samples of Chinese athletes. In their analysis, the factor loading of two items ("My coach tries to motivate me by promising to reward me if I do well" and "My coach expects my whole life to centre on my sport participation") was so low that they had to be deleted. This left 13 items on the scale, apportioned as follows: *controlling use of rewards* (three items), *intimidation* (four items), *negative conditional regard* (four items), and *excessive personal control* (two items).

Zhao et al [15] did not conduct exploratory factor analysis on this scale or discuss whether the dimensions (*controlling use of rewards*, *negative conditional regard*, *excessive personal control*, and *intimidation*) were suitable for athletes in the context of Chinese culture. The present study drew on the work of Bartholomew et al [14] and Zhao et al [15], as well as the characteristics of the local culture, to revise the scale. The results showed that Bartholomew et al.'s [14] structure did not apply to Chinese athletes. Based on the results of exploratory factor analysis, the *intimidation* dimension was deleted in our study. This yielded a scale containing nine items: *controlling use of rewards* (3 items), *negative conditional regard* (3 items), and *excessive personal control* (3 items).

Drawing on Schumann's 7-point guide, we made the following conclusions. (1) Molecular precondition. There was an approximately symmetric (sample-population) distribution (except for the skewed error rate), which made correlations between and across studies likely. (2) Molar precondition (reliability). The relevant performance (rs $\geq$ .837) indices exhibited high test–retest reliability, as demonstrated by the correlation between Study 2 and Study 3, suggesting that the responses to the Controlling Coach Behaviors Scale had reliable stability. (3–4) Convergence and discrimination. All of the samples in Study 2 were correlated with autonomy support, subjective vitality, decreased sense of worth, motivation level, etc. Consistent with previous studies [35], we observed that both negative conditional regard and excessive personal control were related to negative emotions. Inconsistent with previous studies

[35,38], controlling use of reward was positively correlated with positive outcomes such as autonomy support, suggesting that the controlling use of reward has distinct effects in the context of Chinese culture. This is discussed in more detail below (section on "the differences between Eastern and Western athletes' experience of controlling coaching"). (5) Stability. Stability is the degree to which (absolute) scores remain constant from test to retest. In this study, stability refers to the degree to which the structure of the scale does not change with sample types or time. Our results showed that the structure of the scale did not change with sample type (female, male, open sports, and closed sports). However, due to the limitations of the study, we did not measure samples at different times. The stability over time can be explored in future studies. (6) Reproducibility. The subjects were randomly divided into two groups in Study 2. The relationships between the three dimensions and the other variables were consistent in all of the samples, indicating stable relationships. (7) Generalisability. The overall picture of reliability, correlational structure and stability was similar in all of the samples. Together, these seven points suggest that the scale can be further extended.

## Similarities between Eastern and Western athletes' experience of controlling coaching

This study explored the specific characteristics of controlling coach behaviours in a Chinese setting. The analyses showed that the Eastern and Western athletes had similar experiences with controlling coach behaviours, reflecting the structural stability of the two dimensions of excessive personal control and negative conditional regard (see Table 2–4). In educational contexts, excessive personal control and negative conditional attention negatively affected the athletes' emotions and motivation [35]. These two control methods and their effects were found to be consistent across cultural contexts.

In addition to, SDT states that there are two types of motivation: autonomous motivation (the presence of will, initiative, choice, and involvement when engaged in an activity or behaviour) and controlled motivation (in which the reason for engaging in an activity is largely external). Within the framework of SDT, researchers believe that coaching styles have an essential impact on athletes' motivation and subsequent behaviour. The SDT literature provides substantial evidence of a positive correlation between autonomy-supportive coaching styles and autonomous motivation [4,5,36]. The controlling coaching style has been shown to have a statistically significant positive correlation with controlled motivation [4,5,37]. The present study did show a positive and significant correlation between controlling coaching style and controlled motivation, which is consistent with previous SDT-related studies [4,5,37]. Three dimensions of the scale were positively and significantly correlated with amotivation and external regulation (see Table 2–4), both of which are manifestations of controlled motivation and support the external validity of the scale. From an SDT perspective, a controlling coaching style and the resulting maladaptive internalised behaviours make athletes feel that their behaviour is regulated by external factors, leading to increased controlled motivation.

## The differences between Eastern and Western athletes' experience of controlling coaching

The results of this study also showed that Eastern and Western athletes experienced some controlling coaching styles differently. Firstly, Bartholomew et al [14] observed that the controlling coaching style was negatively correlated with autonomy support (*controlling use of rewards*, $r = -.18$; *negative conditional regard*, $r = -.05$; *intimidation*, $r = -.38$; *excessive personal control*, $r = -.36$; and overall, $r = -.46$). However, Zhao et al [15] observed that two of these dimensions,

*controlling use of rewards* (*r* = -.091) and *excessive personal control* (*r* = -.154), were not associated with autonomy support. In fact, in the present study, we found a significant *positive* correlation (*r* = .192) between controlling use of rewards and autonomy support. It is worth noting that the participants targeted by Bartholomew et al [14] and Zhao et al [15] were from different cultural backgrounds, and cultural differences may explain this instability in external validity.

Secondly, it is worth noting that *controlling use of rewards* was positively correlated with introjected regulation, identified regulation, integrated regulation, and intrinsic regulation. Previous evidence from educational settings has suggested that controlling use of rewards has a destructive effect on intrinsic regulation [35,38], and related studies in the field of sports also support this claim [39,40]. However, according to cognitive evaluation theory, intrinsic regulation is not undermined when rewards are considered affirmations of ability rather than control over behaviour. For example, research by Amorose et al [39] showed that the effects of a controlling coaching style were not necessarily negative, and that this style could allow athletes to achieve their goals by increasing their external regulation. The positive correlations that we found between *controlling use of rewards* and identified regulation, integrated regulation, and intrinsic regulation support cognitive evaluation theory. They also demonstrate that in the Chinese cultural context, although rewards do lead to an increase in controlled motivation of athletes, they do not hurt autonomous motivation.

Thirdly, Burnout is a significant negative factor affecting the mental health of athletes [41,42]. According to Raedeke et al [43], emotional fatigue is mainly reflected in a reduced sense of value, a reduced sense of achievement, and emotional and physical exhaustion. In relation to physical activity and exercise, burnout was associated with decreased physical performance and negative behaviours such as abandoning physical activity altogether [44]. Athletes with high levels of burnout typically exhibit behaviours that move away from exercise [45]. Several studies have explored the relationship between coaching behaviour and burnout [46–48]. Castillo et al [47] observed that controlling coaching style was positively and significantly correlated with burnout. In the present study, we found that *negative conditional regard* and *excessive personal control* were positively and significantly correlated with reduced sense of value, reduced sense of accomplishment, and physical exhaustion. However, it is worth noting that *controlling use of rewards* only appeared to reduce athletes' sense of the value in their sport; it was not related to physical exhaustion or reduced sense of achievement.

Fourthly, in our final scale, the *intimidation* dimension was removed because exploratory factor analysis showed that all of the items in this subscale had cross-factor loadings. In the original scale, *intimidation* referred to using shouting, threats, and corporal punishment to humiliate or degrade athletes and control their behaviour. *Intimidation* has been shown to cause athletes to develop higher cognitive and somatic anxiety [49]. Theoretically, in a task-oriented collectivist environment, personal feelings and goals may not be prioritized relative to collective goals. Therefore, when coaches help athletes to pursue collective goals, the athletes may interpret intimidating behaviours as encouragement. That is, in the Chinese cultural context, intimidation may not only be a means of control but also an incentive. In our statistical analysis, most of the items in the intimidation dimension showed cross-factor loadings on *negative conditional regard*. *Negative conditional regard* is responding to athletes' failure with indifference in the hope that the athlete will work harder and achieve better results in the future. Both dimensions represent the coach's rejection of the Chinese athlete. In the context of Chinese culture, coaches' threat to athletes can often be realized through indifference. For example, the punishment in the item "My coach uses the threat of punishment to keep me in line during training" can be realized through neglect and indifference. Many Chinese athletes say, "I would rather a coach scold me than ignore me", so in the Chinese context, indifference to athletes may be the biggest threat. The present study speculates that intimidation items

showed cross-factor load under the *negative conditional regard* subscale, which the particular cultural background of China may cause.

## The correlation structure across contexts

The correlation structure had similarities and differences across cultural contexts but was the same in all educational contexts. For example, negative conditional regard and excessive personal control also had negative effects in educational contexts. Two main methods for teachers to control students are autonomy suppression and conditional regard [50]. Studies have shown that negative conditional regard entails a psychological price [51–54]. In addition, Barber [31] proposed that the use of excessive personal control by parents can compromise children's perceptions of autonomy and undermine their need for relatedness [55,56]. Longitudinal studies have also shown that parent psychological control is positively associated with children's daily self-worth instability [57], and studies have provided strong evidence for correlations between parent conditional regard and failure (viz. competence), contingent shame and contingent self-worth among young people [51,52,58].

We observed that the correlation structures have similarities across sport contexts. In Study 2, we randomly divided the subjects into two groups based on the type of sports. As shown in Table 2–4, the correlations between controlling behaviours and variables such as autonomy support, subjective vitality, diminished senses of worth and motivation were the same in both subsamples.

In conclusion, the present study observed that (1) the dimension of the Controlling Coach Behaviors Scale had its particularity in the Context of Chinese culture. We revised the Controlling Coach Behaviors Scale for use with Chinese athletes. The final scale included three dimensions: *controlling use of rewards*, *negative conditional regard*, and *excessive personal control*; (2) in the context of Chinese culture, *intimidation* and *negative conditional regard* overlapped, and indifference might be one of the important means by which Chinese coaches threaten athletes; (3) although rewards played a controlling role in the Context of Chinese culture, they were not negative. Reward control could awaken an athlete's internal motivation. It was worth noting that, In the present study, we explored a three-factor controlling coaching model (controlling use of rewards, negative conditional regard, and excessive personal control) and a two-factor controlling coaching style model (negative conditional regard and excessive personal control). The fit of both models was good. This suggests that in subsequent studies, researchers can select the scale they wish to use according to their greatest areas of interest. Our external validity tests suggest that the mechanisms underlying the three dimensions (controlling use of rewards, negative condition regard, excessive personal control) might be different, leading to differences in the effects of other variables. Follow-up research should further explore the mechanisms underlying these different dimensions and their effects on other variables. In summary, the present study observed that cultural differences played an important role in the communication between athletes and coaches, so we should combine the national conditions based on learning from the successful cases of other countries. We were only at the preliminary inference stage and did not explore the root causes of differences in the Scale of Controlling Coach Behaviors. Follow-up studies should further explore the root causes of different dimensions of the Controlling Coach Behaviors Scale and the influence of different dimensions on other psychological factors.

## Open practices statement

The data for all experiments is available at https://osf.io/6rwts/.

## Author Contributions

**Funding acquisition:** Daliang Zhao.

**Validation:** Yu Zhou.

**Writing – original draft:** Daliang Zhao, Yu Zhou.

**Writing – review & editing:** Daliang Zhao.

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
