## [Decision Letter · Decision Letter 0]

30 Aug 2022

PONE-D-22-21232Examining the psychometric properties of the Controlling Coach Behaviors Scale in Chinese Elite AthletesPLOS ONE

Dear Dr. Zhou,

Thank you for submitting your manuscript to PLOS ONE. After careful consideration, we feel that it has merit but does not fully meet PLOS ONE’s publication criteria as it currently stands. Therefore, we invite you to submit a revised version of the manuscript that addresses the points raised during the review process. I found one expert reviewer to comment on your work. I also read the manuscript myself. The referee considers your work important overall and suggested some points to consider before publication, mostly statistical issues. Therefore, I would invite you preparing a revision of your work that addresses all concerns together with a cover letter that provides point-by-point replies.

We look forward to receiving your revised manuscript.

Kind regards,

Michael B. Steinborn, PhD

Section Editor

PLOS ONE

Journal Requirements:

" ext-link-type="uri" xlink:type="simple">https://journals.plos.org/plosone/s/file?id=ba62/PLOSOne_formatting_sample_title_authors_affiliations.pdf"

5. We note you have included a table to which you do not refer in the text of your manuscript. Please ensure that you refer to Table 1 in your text; if accepted, production will need this reference to link the reader to the Table.

6. Please include a copy of Table 5 which you refer to in your text on page 14.

Reviewers' comments:

Reviewer's Responses to Questions

**Comments to the Author**

1. Is the manuscript technically sound, and do the data support the conclusions?

Reviewer #1: Partly

2. Has the statistical analysis been performed appropriately and rigorously? 

Reviewer #1: N/A

3. Have the authors made all data underlying the findings in their manuscript fully available?

Reviewer #1: No

4. Is the manuscript presented in an intelligible fashion and written in standard English?

Reviewer #1: Yes

5. Review Comments to the Author

Reviewer #1: This research is interesting. The writing is very good and the the manuscript is well organized. The reader can follow the arguments. I cannot judge whether the research question is important as I am not the particular expert in the field of coaching. Being a professor for personnel psychology and statistics methods, I am concerned with related statistics methods and I will focus on these points accordingly.

comments

## the theoretical classification of coaching styles might be described more clearly. If possible, a table might be introduced where the styles are listed and described in more detail. To name an example what I mean., see the arrangement in this work: doi: 10.3389/fpsyg.2022.867978, see Table 1 and 2.

## the correlation structure presented here very likely depend on the context. This means, it depends what context is addressed (or referred to) in the items of the questionnaire, but also, what is the context where the sample population lives. The question is whether a general structure could be expected theoretically, apart from context differences, or whether different correlation structures are expected in different well-defined context. This point might be addressed a bit more.

## An interesting point would concern cultural differences in coaching styles. Typically, researchers distinguish between western and eastern cultures, and it would be interesting to know whether the concept of coaching style is viewed universal or specific with respect to culture.

## The hypotheses or research questions should be specified in more detail? What is expected on what theoretical grounds, and what are the ancillary variables that could affect the outcome?

## More information should be provided regarding the used questionnaires. Do they mean the same after translation, for example, from english to chinese, or is this aspect examined in more detail?

## Statistics should be presented in tables and in a more systematic way. Measurements are taken on more than one occasion to study test-retest reliability of performance and subjective ratings? Before interpreting the correlation structure, I suggest considering Schumann's 7-point step guide as a general framework (doi: 10.3389/fpsyg.2022.946626, see Table 1). I suggest presenting a table with population parameters (M, SD, skew) of all variables, and the correlation in the MTMM style. It is important to know the test-retest reliability of the measures, also the reliability of the outcome solutions scales would be important to know. The authors are reporting cronbach alpha but it is important to have test-retest reliability on hand, but see my next point

## reliability and cronbach alpha are different concepts. test-retest reliability refers to how precise the measurement is, so it means do we measure the same at different time points? reliability in terms of measurement precision is typically aimed to maximize, because high levels mean highest precision. This is different for cronbach alpha, which refers to the internal consistency of items under a scale. Here, high scores are not always wished because it would mean that a scale measure consists of very similar questions. therefore, cronbach's alpha is typically optimal when it matches the concept (small concept = high alpha, broadband concept = moderate alpha).

## The discussion is a good read. Yet, the interpretation should be reconsidered a bit.

6. PLOS authors have the option to publish the peer review history of their article (what does this mean?). If published, this will include your full peer review and any attached files.

Reviewer #1: No

---

## [Author Response · Author response to Decision Letter 0]

12 Oct 2022

Response to Reviewers

We sincerely thank the editor and the reviewer for their comments and advice, which we have used to improve the manuscript. Below is a point-by-point response to each comment. The line numbers refer to the revised manuscript.

Editor

Comment 1: I found one expert reviewer to comment on your work. I also read the manuscript myself. The referee considers your work important overall and suggested some points to consider before publication, mostly statistical issues. Therefore, I would invite you preparing a revision of your work that addresses all concerns together with a cover letter that provides point-by-point replies.

Response: We have addressed the reviewer’s concerns about our statistical analyses and the interpretation of the effects. We have incorporated the editor’s and reviewer’s constructive suggestions into the revised manuscript, and we believe we have answered the other questions raised by the reviewers.

Reviewer 

Comment 1: the theoretical classification of coaching styles might be described more clearly. If possible, a table might be introduced where the styles are listed and described in more detail. To name an example what I mean., see the arrangement in this work: doi: 10.3389/fpsyg.2022.867978, see Table 1 and 2.

Response: Thank you very much for your support for our manuscript. We have added a table, called Table 1 in the revised manuscript, that summarises coaching styles (p. 3).

Comment 2：the correlation structure presented here very likely depend on the context. This means, it depends what context is addressed (or referred to) in the items of the questionnaire, but also, what is the context where the sample population lives. The question is whether a general structure could be expected theoretically, apart from context differences, or whether different correlation structures are expected in different well-defined context. This point might be addressed a bit more.

Response: Thank you for pointing this out. We have added a discussion of this issue to the revised manuscript. Specifically, we discuss the generalisability and particularity of the correlation structures from three perspectives: different cultural contexts（p. 21, line 402-490）, different educational contexts and different sports contexts (p. 24, lines 491–510).

Comment 3：An interesting point would concern cultural differences in coaching styles. Typically, researchers distinguish between western and eastern cultures, and it would be interesting to know whether the concept of coaching style is viewed universal or specific with respect to culture.

Response: We do not develop culturally specific concepts of coaching style. However, using the original concept of the controlling coaching style, we examine whether Chinese and Western athletes have different responses to the controlling coaching style. We discuss the similarities and differences in the responses of Eastern and Western athletes towards the controlling coaching style on pages 21 to 22 (lines 02-490).

Comment 4：The hypotheses or research questions should be specified in more detail? What is expected on what theoretical grounds, and what are the ancillary variables that could affect the outcome?

Response: Thank you for this comment. We have added a discussion of the relevant assumptions to the revised manuscript (p. 4, lines 59–74).

Comment 5：More information should be provided regarding the used questionnaires. Do they mean the same after translation, for example, from english to chinese, or is this aspect examined in more detail?

Response: Thank you for this comment. We have added more information about the questionnaires in the revised manuscript (pp. 6–7, lines 113–122).

Comment 6：Statistics should be presented in tables and in a more systematic way. Measurements are taken on more than one occasion to study test-retest reliability of performance and subjective ratings? Before interpreting the correlation structure, I suggest considering Schumann's 7-point step guide as a general framework (doi: 10.3389/fpsyg.2022.946626, see Table 1). I suggest presenting a table with population parameters (M, SD, skew) of all variables, and the correlation in the MTMM style. It is important to know the test-retest reliability of the measures, also the reliability of the outcome solutions scales would be important to know. The authors are reporting cronbach alpha but it is important to have test-retest reliability on hand, but see my next point.

Response: We agree with this suggestion and have revised the tables accordingly (pp. 13–19). The study draws on Schumann’s 7-point guide as a general framework (Schumann et al., 2022). In Study 3, we examine the test-retest reliability of the scale (pp. 19–20).

Comment 7：reliability and cronbach alpha are different concepts. test-retest reliability refers to how precise the measurement is, so it means do we measure the same at different time points? reliability in terms of measurement precision is typically aimed to maximize, because high levels mean highest precision. This is different for cronbach alpha, which refers to the internal consistency of items under a scale. Here, high scores are not always wished because it would mean that a scale measure consists of very similar questions. therefore, cronbach's alpha is typically optimal when it matches the concept (small concept = high alpha, broadband concept = moderate alpha).

Response: Thank you for pointing this out. We have added a third study, Study 3, which measures the test-retest reliability of the Controlling Coach Behaviors Scale (pp. 19–20).

Comment 8：The discussion is a good read. Yet, the interpretation should be reconsidered a bit.

Response: Thank you very much for this suggestion. We have reorganised the discussion section to more clearly analyse the similarities and differences between Chinese and Western athletes’ perceptions of controlling coaching.

Journal Requirements

Comment 1: Please ensure that your manuscript meets PLOS ONE's style requirements, including those for file naming. The PLOS ONE style templates can be found at

https://journals.plos.org/plosone/s/file?id=ba62/PLOSOne_formatting_sample_title_authors_affiliations.pdf"

Response: Thank you for these links. We have modified the manuscript using the PLOS One style templates.

 Comment 2: Please provide additional details regarding participant consent. In the ethics statement in the Methods and online submission information, please ensure that you have specified what type you obtained (for instance, written or verbal, and if verbal, how it was documented and witnessed). If your study included minors, state whether you obtained consent from parents or guardians. If the need for consent was waived by the ethics committee, please include this information.

Response: We have added information about participant consent (p. 7, lines 125–126). All of the procedures performed in this study were approved by the Institutional Review Board of the Guangzhou Sport University. In all three studies, all of the participants or their parents signed informed consent forms.

Comment 3: PLOS requires an ORCID iD for the corresponding author in Editorial Manager on papers submitted after December 6th, 2016. Please ensure that you have an ORCID iD and that it is validated in Editorial Manager. To do this, go to ‘Update my Information’ (in the upper left-hand corner of the main menu), and click on the Fetch/Validate link next to the ORCID field. This will take you to the ORCID site and allow you to create a new iD or authenticate a pre-existing iD in Editorial Manager. Please see the following video for instructions on linking an ORCID iD to your Editorial Manager account: https://www.youtube.com/watch?v=_xcclfuvtxQ

Response: We have valid ORCID iDs, as required: https://orcid.org/0000-0002-2054-6911

Comment 4: Your ethics statement should only appear in the Methods section of your manuscript. If your ethics statement is written in any section besides the Methods, Comment 4please move it to the Methods section and delete it from any other section. Please ensure that your ethics statement is included in your manuscript, as the ethics statement entered into the online submission form will not be published alongside your manuscript.

Response: Thank you for this information. We have moved the ethics statement to the Methods section (p. 7, lines 126–129).

Comment 5: We note you have included a table to which you do not refer in the text of your manuscript. Please ensure that you refer to Table 1 in your text; if accepted, production will need this reference to link the reader to the Table.

Response: We have renamed the tables in this revision and have ensured that each one is referred to in the text (for example, p.3, lines 35).

Comment 6: Please include a copy of Table 5 which you refer to in your text on page Response: We have renamed the tables in this revision, and we have ensured that each one is referred to in the text (for example, p.3, lines 35).

Comment 7: The //requires authors to make all data underlying the findings described in their manuscript fully available without restriction, with rare exception (please refer to the Data Availability Statement in the manuscript PDF file). The data should be provided as part of the manuscript or its supporting information, or deposited to a public repository. For example, in addition to summary statistics, the data points behind means, medians and variance measures should be available. If there are restrictions on publicly sharing data—e.g. participant privacy or use of data from a third party—those must be specified.

Response: The data used in our experiments are available at https://osf.io/6rwts/. We have also added a table (Table 2-1 in the new version) that provides the means and standard deviations of the variables.

---

## [Decision Letter · Decision Letter 1]

31 Oct 2022

PONE-D-22-21232R1Examining the psychometric properties of the Controlling Coach Behaviors Scale in Chinese Elite AthletesPLOS ONE

Dear Dr. Zhou,

Thank you for submitting your manuscript to PLOS ONE. After careful consideration, we feel that it has merit but does not fully meet PLOS ONE’s publication criteria as it currently stands. Therefore, we invite you to submit a revised version of the manuscript that addresses the points raised during the review process. Editor's comment. As you can see from the review, R1 is satisfied and has only some very minor remarks. Based on my own reading, the manuscript is ready for publication and makes a good contribution to the field. I suggest preparing the final version of the manuscript and I will formally accept the paper in the next submission round. Please also check the reference list for completeness (e.g., doi number, page information, etc.). Also, Tables should be presented in APA style (e.g., headers, avoid vertical lines, etc.) Please submit your revised manuscript by Dec 15 2022 11:59PM. If you will need more time than this to complete your revisions, please reply to this message or contact the journal office at plosone@plos.org. Please include the following items when submitting your revised manuscript:A rebuttal letter that responds to each point raised by the academic editor and reviewer(s). You should upload this letter as a separate file labeled 'Response to Reviewers'.A marked-up copy of your manuscript that highlights changes made to the original version. You should upload this as a separate file labeled 'Revised Manuscript with Track Changes'.An unmarked version of your revised paper without tracked changes. You should upload this as a separate file labeled 'Manuscript'.If applicable, we recommend that you deposit your laboratory protocols in protocols.io to enhance the reproducibility of your results. Protocols.io assigns your protocol its own identifier (DOI) so that it can be cited independently in the future. For instructions see: https://journals.plos.org/plosone/s/submission-guidelines#loc-laboratory-protocols. Additionally, PLOS ONE offers an option for publishing peer-reviewed Lab Protocol articles, which describe protocols hosted on protocols.io. Read more information on sharing protocols at https://plos.org/protocols?utm_medium=editorial-emailutm_source=authorlettersutm_campaign=protocols.

We look forward to receiving your revised manuscript.

Kind regards,

Michael B. Steinborn, PhD

Section Editor

PLOS ONE

Journal Requirements:

Reviewers' comments:

Reviewer's Responses to Questions

**Comments to the Author**

1. If the authors have adequately addressed your comments raised in a previous round of review and you feel that this manuscript is now acceptable for publication, you may indicate that here to bypass the “Comments to the Author” section, enter your conflict of interest statement in the “Confidential to Editor” section, and submit your "Accept" recommendation.

Reviewer #1: All comments have been addressed

2. Is the manuscript technically sound, and do the data support the conclusions?

Reviewer #1: Yes

3. Has the statistical analysis been performed appropriately and rigorously? 

Reviewer #1: Yes

4. Have the authors made all data underlying the findings in their manuscript fully available?

Reviewer #1: Yes

5. Is the manuscript presented in an intelligible fashion and written in standard English?

Reviewer #1: Yes

6. Review Comments to the Author

Reviewer #1: The manuscript has improved considerably and is in good shape and a good read overall. It is ready for publication and I have only a few very minor comments (see below).

168-169

"...the study draws on Schumann’s 7-point guide as a general framework (Schumann et al., 2022)...."

At this occasion, I recommend adding a few more information, for example, in this way: The study draws on Schumann’s 7-point guide as a general framework (Schumann et al., 2022). Accordingly, molecular (skew check) and molar (reliability) preconditions are basically fulfilled. The questionnaires has been validated by previous research as sufficiently reliable (study) and also discriminates from other non-related concepts.

discussion

the discussion is very good! It is informative and insightful. Maybe the points could 1-7 could made more explicit. For example, at the occasion "-point 5- stability", it could be stated first what is the requirement in the ideal norm, and in the definition, and then discussing the issue further (what is the reality in the present case).

overall, very good work!

7. PLOS authors have the option to publish the peer review history of their article (what does this mean?). If published, this will include your full peer review and any attached files.

Reviewer #1: No

---

## [Author Response · Author response to Decision Letter 1]

6 Nov 2022

Response to Reviewers

We sincerely thank the editor and the reviewer for their comments and advice, which we have used to improve the manuscript. Below is a point-by-point response to each comment. The line numbers refer to the revised manuscript.

Editor

Comment 1: As you can see from the review, R1 is satisfied and has only some very minor remarks. Based on my own reading, the manuscript is ready for publication and makes a good contribution to the field. I suggest preparing the final version of the manuscript and I will formally accept the paper in the next submission round. Please also check the reference list for completeness (e.g., doi number, page information, etc.). Also, Tables should be presented in APA style (e.g., headers, avoid vertical lines, etc.)

Response: Thank you very much for your affirmation of our manuscript. We have carefully checked the completeness of the reference list and corrected the writing style according to the requirements of the journal. No retracted papers are cited in the revised manuscript. In addition, we have reformatted the tables according to APA style guidelines.

Reviewer 

Comment 1: The manuscript has improved considerably and is in good shape and a good read overall. It is ready for publication and I have only a few very minor comments (see below).

168-169 "...the study draws on Schumann’s 7-point guide as a general framework (Schumann et al., 2022)...."At this occasion, I recommend adding a few more information, for example, in this way: The study draws on Schumann’s 7-point guide as a general framework (Schumann et al., 2022). Accordingly, molecular (skew check) and molar (reliability) preconditions are basically fulfilled. The questionnaires have been validated by previous research as sufficiently reliable (study) and also discriminates from other non-related concepts.

Response: Thank you very much for your support. We have added more information to the revised manuscript (pp. 10, 169–172).

Comment 2: the discussion is very good! It is informative and insightful. Maybe the points could 1-7 could made more explicit. For example, at the occasion "-point 5- stability", it could be stated first what is the requirement in the ideal norm, and in the definition, and then discussing the issue further (what is the reality in the present case).

Response: Thank you for pointing this out. We have expanded the discussion on Points 1–7 (pp. 21, 401–422).

---

## [Editor Report · Decision Letter 2]

8 Nov 2022

Examining the psychometric properties of the Controlling Coach Behaviors Scale in Chinese Elite Athletes

PONE-D-22-21232R2

Dear Dr. Zhou,

We’re pleased to inform you that your manuscript has been judged scientifically suitable for publication and will be formally accepted for publication once it meets all outstanding technical requirements.

Editor's comment. I read the final version of the manuscript again and found, in agreement with R1, it is a really good work, insightful debated and the arguments are clear. 太好了！

Kind regards,

Michael B. Steinborn, PhD

Section Editor

PLOS ONE
---

## [Editor Report · Acceptance letter]

14 Nov 2022

PONE-D-22-21232R2 

Examining the psychometric properties of the Controlling Coach Behaviors Scale in Chinese Elite Athletes 

Dear Dr. Zhou:

I'm pleased to inform you that your manuscript has been deemed suitable for publication in PLOS ONE. Congratulations! Your manuscript is now with our production department. 

Kind regards, 

on behalf of

Dr. Michael B. Steinborn 

Section Editor

PLOS ONE